# Entity Disambiguation on a Tight Labeling Budget

**Audi Primadhanty** and **Ariadna Quattoni**
Universitat Politècnica de Catalunya / Barcelona, Spain
audi.primadhanty@upc.edu, aquattoni@cs.upc.edu

## Abstract

Many real-world NLP applications face the challenge of training an entity disambiguation model for a specific domain with a small labeling budget. In this setting there is often access to a large unlabeled pool of documents. It is then natural to ask the question: which samples should be selected for annotation? In this paper we propose a solution that combines feature diversity with low rank correction. Our sampling strategy is formulated in the context of bilinear tensor models. Our experiments show that the proposed approach can significantly reduce the amount of labeled data necessary to achieve a given performance.

## 1 Introduction

Many real-world NLP applications face the challenge of training an entity disambiguation model for a specific domain with a small labeling budget. For example, a defense research analyst might be interested in mapping mentions of military equipment appearing in military research articles to a target knowledge base describing emergent defense technologies. The time of an expert might be costly and we would like to minimize the amount of annotations required from them.

When training an entity disambiguation model one has often access to a large unlabeled pool of documents. We assume that we can recognize mentions of the target class and that we can generate a set of candidate Knowledge Base (KB) entities for each of them. Solving the entity disambiguation task involves training a model that takes a target mention and a set of candidate KB entities as inputs and outputs the correct grounding (i.e. the correct entity in the KB). We assume that we have a small annotation budget and that our goal is to train the best model possible under the budget constraints.

One natural approach to tackle this problem is to design sampling strategies that attempt to maximize the diversity of the selected samples. We

follow this line of work and develop a method that combines a model correction step designed to improve the generalization performance of models learnt under tight annotation budgets with a diversity sampling strategy.

Our work is framed in the context of bilinear tensor models for entity disambiguation. These types of models score mention-entity pairs by exploiting a rich representation of the mention context and entity descriptions in the target KB. Furthermore, our model has a bilinear lexicalized attention mechanism that assigns a score to every pair in the Cartesian product space of context and entity description words. Our first observation is that the lexicalized bilinear attention matrix will be typically low-rank. Following this insight we develop a model correction technique based on matrix completion that can improve the generalization performance of bilinear tensor models trained on tight annotation budgets. We combine this with feature diversity sampling, and it is by combining these two ideas that we obtain optimal results.

The two main contributions of this work are: 1) We develop a novel strategy to improve the generalization performance of bilinear tensor models trained under tight annotation budgets. Our strategy combines low-rank matrix completion of a lexical attention matrix and feature diversity sampling. 2) Our experiments on entity disambiguation show that the proposed approach can significantly reduce the amount of labeled data necessary to achieve a given entity disambiguation performance.

## 2 Related Work

The sampling problem in learning under a budget is related to active learning (Zhang et al., 2022). A few works addressed active learning for entity linking (Lo and Lim, 2020; Oberhauser et al., 2020). Different to active learning in the low budget setting, we don't assume the existence of a prior model and thus our setting resembles the cold start prob-

lem (Yuan et al., 2020; Jin et al., 2022). To the best of our knowledge we are the first ones to address cold start for entity disambiguation. As in active learning, the sampling approach can exploit either the informativeness or the representativeness of data points (Zhang et al., 2022). Hacohen et al. (2022) showed that, for low-budget scenarios, representativeness is a better query strategy. We propose two representative-based sampling approaches and study their effectiveness for the case of entity disambiguation. Our method combines diversity sampling with low-rank matrix completion, which has been shown to help generalization in low-budget scenarios (Primadhanty et al., 2015; Quattoni et al., 2014).

## 3 Tensor Model with Bilinear Attention

### 3.1 Learning Setting

Our goal is to train a local entity disambiguation model that takes as input an entity mention in context and a set of candidate KB entities and predicts the gold entity in the KB to which the mention should be linked. More formally, we assume that we are given: 1) A set of target entities $\mathcal{KB} = \{e_1, e_2, \ldots\}$, each with an associated description. 2) A training set tuples $(m, mc, C(m), g(m)) \in \mathcal{T}$, where $m$ is an entity mention, $mc$ is the mention context, $C(m)$ is a set of candidate entities (i.e. a subset of $\mathcal{KB}$) generated by some candidate generation algorithm, and $g(m)$ is the correct entity for the given mention[1]. Our goal is to use $\mathcal{T}$ and $\mathcal{KB}$ to train a model that takes as input a tuple $(m, mc, C(m)$ and predicts an entity[2] $e \in \mathcal{KB}$.

In order to train our model, we also assume that we can compute a set of features of the mention, its context and all entities in the KB. More precisely, for each mention, we compute (contextual) pre-trained BERT embedding vectors of the mention text $\mathbf{b}_m$ and its context $\mathbf{b}_{mc}$, as well as a sparse feature vector of the context $s(mc)$. On the KB side, we represent each entity by a pre-trained BERT embedding $\mathbf{b}_e$ and a sparse feature function $s(e)$ of its description. The embeddings $\mathbf{b}_{mc}$ and $\mathbf{b}_e$ are computed by taking the average of the BERT embeddings of all words in $mc$ and the entity $e$ description, respectively. The functions $s(mc)$ and

$s(e)$ return the set of their unique words[3].

In a learning under a budget setting we assume that we are given an unlabeled dataset of tuples $(m, mc, C(m)) \in \mathcal{U}$ and a labeling budget $n$. We can select $n$ samples from $\mathcal{U}$ to be labeled to create the training set $\mathcal{T}$. The goal is to train the most accurate model under the given budget constraint.

### 3.2 Tensor Entity Disambiguation Model

We will consider tensor models of the form:

$$P(e|(m, mc, C(m)) = \frac{exp^{\theta(e, mc, C(m))}}{\sum\limits_{e' \in C(m)} exp^{\theta(e', mc, C(m))}}$$

The scoring function $\theta(e, m, mc)$ computes the compatibility between an entity and a mention in context. This function will be parameterized by a $d_{mc} \times d_e$ attention matrix $\mathbf{A}$, where $d_{mc}$ is the number of unique words appearing in the context of any mention and $d_e$ is the number of unique words appearing in the description of any entity. The attention matrix is indexed by context mention words and entity description words and the $(i, j)$ matrix cell is expected to capture the compatibility between a mention context word and an entity description word. The scoring function is also parameterized by two $d_{\mathbf{b}} \times d_{\mathbf{b}}$ matrices $\mathbf{W}$ and $\mathbf{Z}$, where $d_{\mathbf{b}}$ is the dimensionality of the BERT embeddings. This two matrices are expected to capture the compatibility between mention and entity description embeddings and between mention context and entity description embeddings, respectively. The scoring function is the sum of a contextual and mention term:

$$\theta(e, m, mc) = \sum\limits_{p \in s(mc), q \in s(e)} \mathbf{A}[p, q] + \mathbf{b}_m^t \mathbf{W} \mathbf{b}_e + \mathbf{b}_{mc}^t \mathbf{Z} \mathbf{b}_e$$

To train a model we perform standard max-likelihood estimation and find the parameter matrices $\mathbf{A}$, $\mathbf{W}$ and $\mathbf{Z}$ that minimize the negative log-likelihood of the examples in $\mathcal{T}$.

## 4 Diversity Sampling and Low-rank Matrix Completion Correction

In this section we present our approach for learning an entity disambiguation under a budget setting. Our approach combines a low-rank model correction step with diversity sampling.

---

[1] For training we assume that $g(m) \in C(m)$, otherwise we manually add it to the set returned by $C(m)$.

[2] At test time we cannot assume that the correct entity for a given mention will be in the candidate set, in practice however we work with candidate generation algorithms of high recall.

[3] After removing stop words.

## 4.1 Matrix Completion Correction

The intuition behind our model correction strategy is quite simple. Think of the attention matrix $\mathbf{A}$ as an instance of the classical collaborative filtering problem. In this problem one assumes that some users (rows of the matrix) have rated a set of movies (columns of the matrix) and the goal is to predict the ratings of users on movies that they have not rated. The underlying assumption behind the matrix completion solution to this problem is that, to predict unknown ratings for a given user, we can interpolate the predictions of similar users (i.e. users that made similar ratings). The low-rank observation stems from the realization that there might be a few users (i.e. rows) and a few movies (i.e. columns) that can serve as a basis to make all predictions, i.e. unknown predictions can be expressed as linear interpolations of these basis vectors. In the case of the model's attention matrix $\mathbf{A}$, rows correspond to mention contextual words and columns to entity description words. The low-rank basis assumption states that there is a subset of context and entity words from which one can guess the compatibility of all other word pairs by performing low-rank matrix completion.

More precisely, the model correction algorithm that we propose is as follows: Given a set $\mathcal{R}$ of $l$ rank value parameters for the matrix completion, we compute the $max(\mathcal{R})$-rank singular value decomposition of $\mathbf{A}$. Then, for each $r_i \in \mathcal{R}$, we build the rank-$r_i$ low-rank reconstruction, keep only the positive values and convert it into binary matrix $\mathbf{A}_{r_i}$, and we compute their mean $\mathbf{A_m}$. Next, for each entry $\mathbf{A}(i,j)$ we compute its reconstruction uncertainty: $\mathbf{A_u}(i,j) = \text{Var}([\mathbf{A}_{r_1}(i,j) \dots \mathbf{A}_{r_l}(i,j)])$. Finally, we create the complete matrix $\mathbf{A}^* = \mathbf{A_m} - \mathbf{A_u}$ and add bias; given some threshold $th$, if $\mathbf{A}^*(i,j) < th$ then $\mathbf{A}^*(i,j) = th$.

## 4.2 Diversity Sampling for Optimal Attention Completion

The model correction step described in the previous section exploits the information in the observed entries of $\mathbf{A}$ in order to predict the unseen entries. To guarantee the success of the completion step we must ensure that the observed entries in $\mathbf{A}$ provide sufficient information to produce a good estimate (Ruchansky et al., 2015). The weights in $\mathbf{A}$ correspond to features in the Cartesian product space of mention and entity sparse word features. Under a budget constraint we want to select a set of $n$ samples $(m, mc, C(m) \in \mathcal{U}$ that gives us the most information to produce a good completion of $\mathbf{A}$. To achieve this we propose to select a sample that maximizes the coverage of the features in the Cartesian product space. We call this sampling strategy: **cross-product diversity**.

The **cross-product diversity** sampling problem can be reduced to an instance of a combinatorial N-set-cover problem. In the classical N-set cover problem, one is given a set of elements $\mathcal{P} = \{p_1, p_2, \dots\}$ (called the universe), a collection $\mathcal{S}$ of $k$ subsets of $\mathcal{P}$ and a budget $N$ on the number of subsets that can be used for the cover. The problem consists in identifying a sub-collection of $N$ subsets of $\mathcal{S}$ whose union maximizes the number of elements of $\mathcal{P}$ covered. While this problem is known to be NP-hard, there exists several approximation algorithms, we use a simple greedy algorithm. In our case the universe will be the set of features in the Cartesian-product space and the subsets are the samples in the unlabeled pool. Each sample $(m, mc, C(m))$ can be thought as a set of features obtained by taking the union of the cross features with each candidate. More formally, the subset corresponding to sample $(m, mc, C(m)) \in \mathcal{U}$ will be defined as: $\bigcup_{e \in C(m)} s(m, mc) \times s(e)$.

We observed that considering all the features in the Cartesian-product might not be optimal and that it is beneficial to filter out very infrequent features. Therefore we include a feature frequency threshold so that only features that appear more than the allowed threshold are considered. The experiments section provides more details about how this parameter is validated.

The **cross-product diversity** strategy attempts to explicitly maximize diversity in the sparse Cartesian product space. An alternative is to design a sampling strategy that exploits the dense representations, we refer to this strategy as: **dense diversity**. For this strategy we first compute the BERT representation $\mathbf{b}_e$ of all entities appearing in $\mathcal{U}$ as well as the contextual BERT representation $\mathbf{b}_{mc}$ of all mentions in $\mathcal{U}$. Then for each tuple $(m, mc, C(m))$ we generate a vector representation by summing together the $\mathbf{b}_{mc}$ and the average $\mathbf{b}_e$ vectors of all candidate entities in $e \in C(m)$. To select $n$ samples from $\mathcal{U}$ we perform k-means clustering and obtain $n$ clusters, then for each cluster we select the sample closest to the cluster centroid. Compared to the **cross-product diversity** strategy, the **dense**

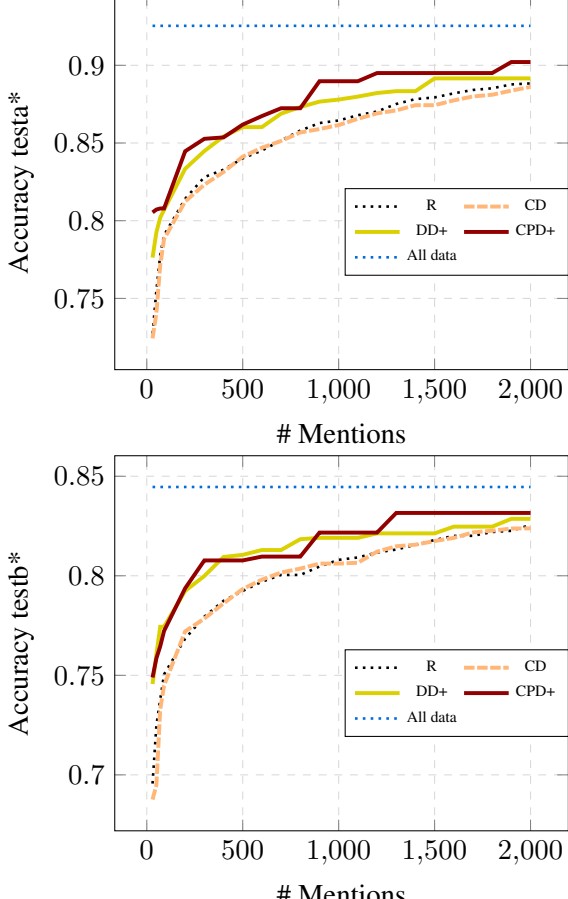

Figure 1: Performance on AIDA testa* and testb*

In addition to the Cross-Product Diversity (**CPD**) and Dense Diversity (**DD**) methods, we experiment with two other baseline methods. Random (**R**) sampling and Candidate Diversity (**CD**) which tries to cover as many unique candidate entities as possible (see section 4.2 for the approach to the N-set-cover problem.).

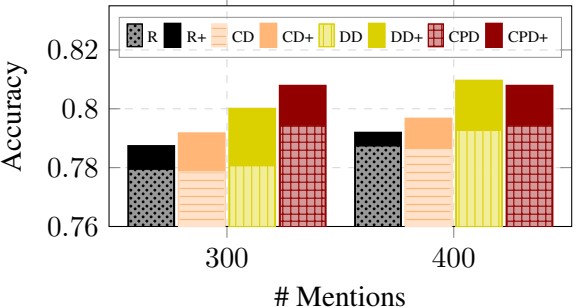

Figure 2: Performance on AIDA testb*.

Figure 1 shows performance on testa* and testb* partition. The methods marked with "+" use low-rank matrix completion correction. We can see that CPD and DD sampling combined with low-rank matrix completion help increases model accuracy compared to random and CD sampling, especially with lower budgets. To give some concrete examples, on testa* CPD+ model correction achieves 81% accuracy with 50 labeled mentions, to get the same accuracy with random sampling you need at least 150 mentions, that is 3 times the number of samples. With 200 mentions CPD+ model correction attains 84.4% accuracy, to get that accuracy with random sampling we need more than 500 annotations. In both instances with our approach the amount of supervision required to attain a given performance is reduced by at least half.

Figure 5, shows two arbitrary points from the learning curve and decouple the sampling strategies from the low-rank correction. Both CPD and DD sampling helps the model's performance even without low-rank matrix completion. Moreover, low-rank matrix completion increase models' accuracies regardless of the sampling strategies.

**diversity** strategy is significantly more costly since it requires running BERT over all mentions and contexts in the unlabeled pool.

## 5 Experiments: Learning with a tight budget

We run our experiments on AIDA CoNLL-YAGO[4] (Hoffart et al., 2011) annotated with entities from WikiData[5]. We follow van Hulst et al. (2020) to select the candidate entities and ignore samples annotated with out-of-KB entities. We produce a learning curve by training models with increasing annotation budgets. Each point uses five random seeds, and we report their average performance. We validate all hyper-parameters (both for the sampling strategy and model optimization) on randomly selected 50 samples from each testa and testb partitions. We will use **testa*** and **testb*** to refer to the testa and testb set that excludes the 50 samples used for validation.

## 6 Conclusion

We have shown that by combining low-rank matrix completion of a lexicalized attention matrix with diversity sampling we can significantly improve the generalization performance of a bilinear entity disambiguation model under tight annotation budget constraints.

---

[4]https://resources.mpi-inf.mpg.de/yago-naga/aida
[5]https://www.wikidata.org

## Limitations

As mentioned in section 3.1, we are focusing on the task of local entity disambiguation, that is, our model is trained to disambiguate entities in isolation. One could further improve the local disambiguation model by imposing some global consistency constraints that capture the interaction between entities but this is outside the scope of this work.

Moreover, in entity disambiguation, we are assuming that the mentions have been correctly detected in the text. It will be interesting to consider how to adapt the methods to the task of end-to-end Entity Linking, where we also need to identify the mentions.

In section 2, we mention how learning under budgets can be related to active learning. We do not try to use our methods as a cold-start technique in active learning in this study, but this can be considered for future works.

## Acknowledgements

This project has received funding from the European Research Council (ERC) under the European Union's Horizon 2020 research and innovation programme under grant agreement No 853459. The authors gratefully acknowledge the computer resources at ARTEMISA, funded by the European Union ERDF and Comunitat Valenciana as well as the technical support provided by the Instituto de Física Corpuscular, IFIC (CSIC-UV). This research is supported by a recognition 2021SGR-Cat (01266 LQMC) from AGAUR (Generalitat de Catalunya).

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

## A Additional Details

### A.1 Model Description

In figure 3, we show an example of an entity disambiguation problem and the source of features and representations that we use in our model:

- $\mathbf{b}(\mathbf{mc})$ is obtained by averaging the BERT embeddings of the context of the mention,

- $\mathbf{b}(\mathbf{m})$ is the BERT embedding of the mention itself,

- $\mathbf{b}(\mathbf{E})$ is the average of BERT embeddings of the entity's description

In addition to that, we also use $\mathbf{s}(\mathbf{mc})$ and $\mathbf{s}(\mathbf{E})$ which can be considered as bag-of-words features of the mention context and entity description.

We also show an example of the learned matrix entry $\mathbf{A}(\mathbf{i}, \mathbf{j})$ that is expected to capture the compatibility between a mention context word and an entity description word.

### A.2 Detailed Model Correction Algorithm

Algorithm 1 describes in more details the low rank (LR) attention completion algorithm we briefly described in section 4.1.

## B Additional Results

Figure 4 shows the same evaluation as Figure 5 done on testa* partition. These results show the same observation as seen in evaluation on testb* partition; the Cross-Product Diversity and Dense Diversity sampling, as well as low rank completion correction can help in tight-budget scenarios.

Figure 5 and 6 shows the breakdown as seen in Figure 5 and 4 for all points in the learning curve, and as we can see, the same observation still holds throughout the curve.

## C Dataset

We run our experiments on AIDA CoNLL-YAGO[6] (Hoffart et al., 2011). The dataset is partitioned into three sets: train, testa and testb. We follow van Hulst et al. (2020) to select the candidate entities and ignore samples annotated with out-of-KB entities. Using this strategies, we obtain 17174, 4422, and 4226 number of annotated mentions for the train, testa and testb partition, respectively.

Originally, testa is a validation partition, and testb is a test partition. But in our scenario, it

---

[6]https://resources.mpi-inf.mpg.de/yago-naga/aida

---

**Algorithm 1:** LR Attention Completion

**Input:**

- $\mathbf{A}$ , Attention matrix.

- $\mathcal{R}$ , a set of $l$ rank value parameters for the matrix completion.

- $th$, bias threshold.

**Output:** $\mathbf{A}^*$.

- Compute the $max(\mathcal{R})$-rank singular value decomposition of $\mathbf{A} = \mathbf{USV}$

- For each $r \in \mathcal{R}$ build the rank-$r$ low-rank reconstruction:
  $\mathbf{A}'_r = U(:, 1:r)S(1:r, 1:r)V(1:r, :)$
  $$\mathbf{A}_r(i,j) = \begin{cases} 1 & \text{if } \mathbf{A}'_r(i,j) > 0, \\ 0 & \text{otherwise.} \end{cases}$$

- Compute the mean reconstruction:
  $$\mathbf{A_m} = \frac{\sum_{i \in \mathcal{R}} \mathbf{A}_{r_i}}{|\mathcal{R}|}$$

- For each entry $\mathbf{A}(i,j)$ compute its reconstruction uncertainty:
  $$\mathbf{A_u}(i,j) = \mathrm{Var}([\mathbf{A}_{r_1}(i,j) \ldots \mathbf{A}_{r_l}(i,j)])$$

- Create the completed matrix:
  $\mathbf{A}' = \mathbf{A_m} - \mathbf{A_u}$
  $$\mathbf{A}^*(i,j) = \begin{cases} \mathbf{A}'(i,j) & \text{if } \mathbf{A}'(i,j) > th, \\ th & \text{otherwise.} \end{cases}$$

- **Return:** $\mathbf{A}^*$

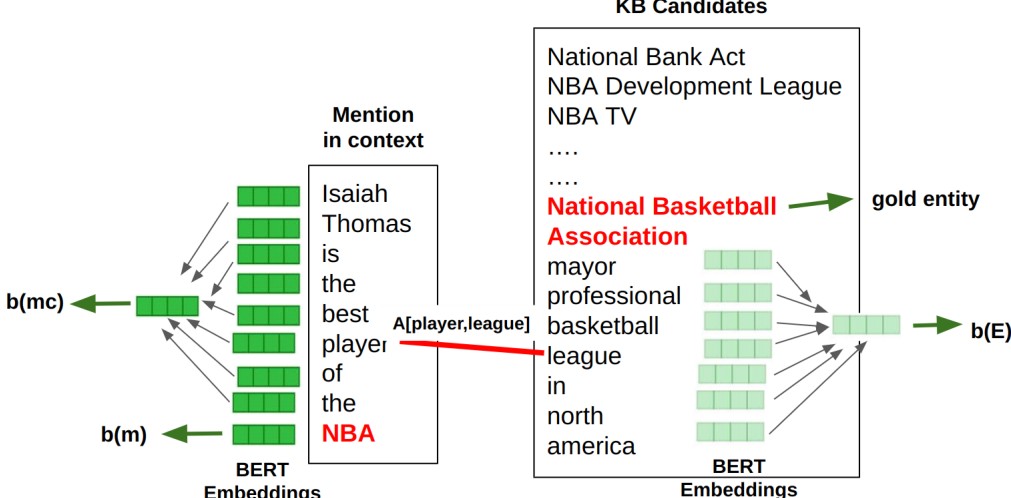

Figure 3: This figure displays a concrete disambiguation training example, where the mention is NBA, and the gold KB entity is National Basketball Association.

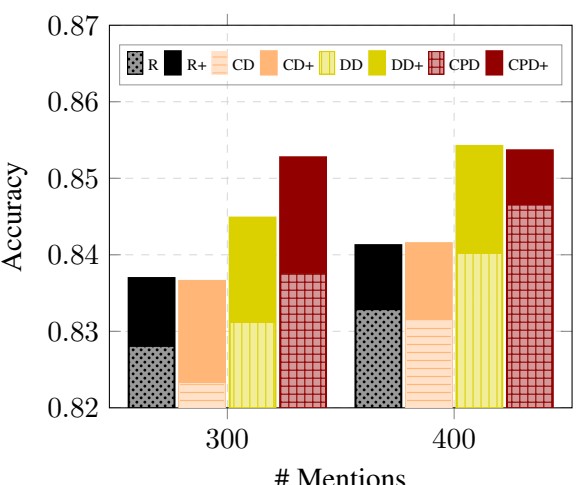

Figure 4: Performance on AIDA testa*.

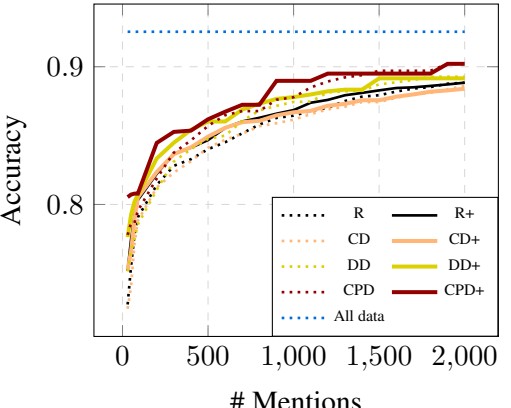

Figure 5: Performance on AIDA testa*.

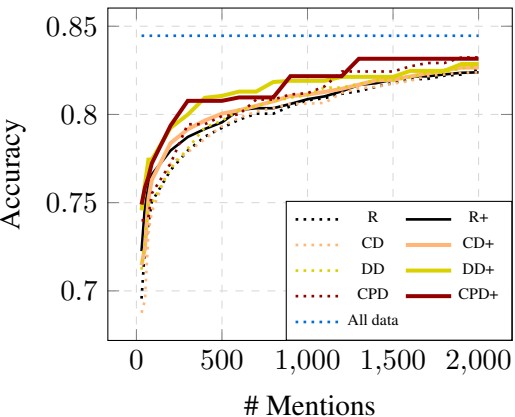

Figure 6: Performance on AIDA testb*.

is unrealistic to assume that we have access to a large validation partition since we only have a small budget for annotation. We create a small validation set from both testa and testb instead, selecting 50 random examples from each. It has also been well documented in previous studies that due to some difference in the distribution, testa seems to be an "easier" partition (models trained on the standard train set will perform better in testa) than testb. So, we test on both to see if the strategies will behave differently depending on how "hard" the data is.

## D Qualitative Analysis

To understand better how the matrix completion correction might help improve performance, we performed a qualitative analysis of our results. We use the model trained on 300 random training sam-

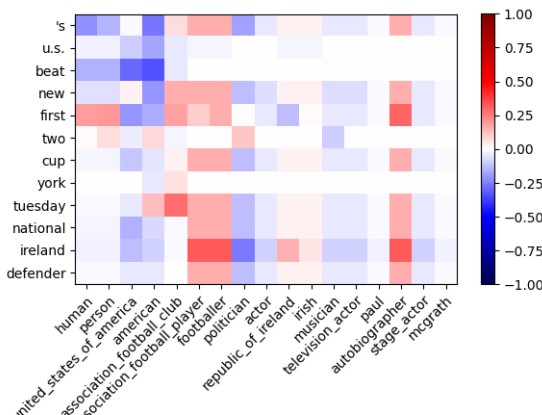

(a) Before matrix completion correction.

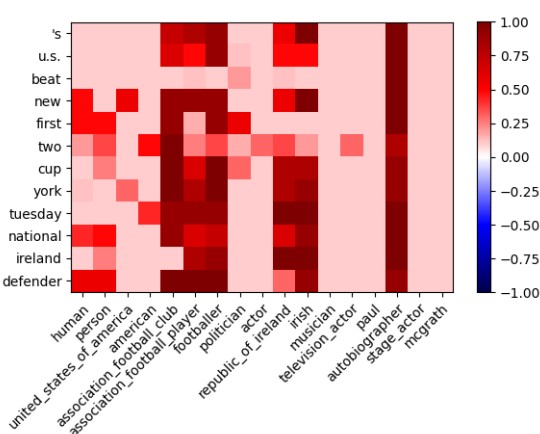

(b) After matrix completion correction.

Figure 7: Heatmap of a subset of the attention matrix before and after matrix completion correction. The model was trained using 300 random mentions. The rows are features of the mentions, while the columns are features of the entities.

ples and compare the correctly identified test samples before and after applying matrix completion correction (i.e. Rand R+). We focus on correctly identified entities not in the training set and choose one, in this case, "Juventus F.C."; two mentions of "Juventus F.C." are identified after applying matrix completion correction. We then look at the feature cross products of one mention and the entity and compare their weights in Rand R+, which shows that R+ can propagate weight to 35 unseen feature products, one being "defender" vs "association_football_club". In 7a, we select a subset of the cross-product space to illustrate how the weight "propagation" works similarly to collaborative filtering mentioned in section 4.1. If we treat the row as users and columns as movies, then let's look at the user "defender" and movie "association_football_club". We can see that the user "defender" has quite similar patterns to many other users such as "national", "tuesday", "cup", and "new". But we don't know if this user will like the movie "association_football_club", so we can use the clue from user "Tuesday", "cup", and "new" to guess that this user will like it. We can see in Figure 7b that the matrix completion correction does that and "propagates" the weight to the cell.

We can also see from the patterns of seen cross products that "association_football_club" is quite similar to "association_football_player" and "footballer", it seems natural that "defender" would also be compatible with "association_football_club". This illustrates how the matrix completion correction step is implicitly learning latent classes of features. In this example, "association_football_club", "association_football_player" and "footballer" seem to belong to the same class of features.

# E Hyperparameters

Here we specify the different hyperparameters we use to obtain the results shown in the paper. We run each experiments with 5 random seeds {1,2,3,4,5} and report the average performance. The learning curve is created using the following number of samples: {10, 30, 50, 70, 90, 200, 300, 400, 500, 600, 700, 800, 900, 1000, 1100, 1200, 1300, 1400, 1500, 1600, 1700, 1800, 1900, 2000}. We use UN-LocBoX (Perraudin et al., 2016) for optimization and uses LIP = 10, TAU = 0.01, and maximum iteration of 30. We use $th = 0.1$ and $\mathcal{R} = \{1 \dots 20\}$.