# OpenReview forum: "Entity Disambiguation on a Tight Labeling Budget"
_EMNLP/2023/Conference — EMNLP 2023 Findings_

### Official Review · Reviewer_XGHN · 2023-07-30

**Soundness:** 3

**Excitement:**

3: Ambivalent: It has merits (e.g., it reports state-of-the-art results, the idea is nice), but there are key weaknesses (e.g., it describes incremental work), and it can significantly benefit from another round of revision. However, I won't object to accepting it if my co-reviewers champion it.

**Paper Topic And Main Contributions:**

The paper presents a study of different data sampling strategies for an entity disambiguation model. The approached task is a simplified entity linking task where the mentions are assumed to have been previously identified. The authors additionally experiment with a low-rank matrix completion method applied to a mention context to entity descriptor attention matrix part of the objective function of the model.

**Questions For The Authors:**

A. Would be good to see -- certainly it would strengthen the work-- how the discussed data sampling approaches perform across multiple models and datasets. Did you do any exploration to this end?
B. For dense diversity you used k-means clustering to end up selecting for each cluster the sample closest to the centroids. Why not use k-medoids for this task? As opposed to k-means the cluster centres are actual data points from the data, and you can specify any dissimilarity measure, i.e, you are not limited to the Euclidian distance
C. Why did you settle on that particular entity disambiguation model? Would be good to see some references if that is the standard approach in the field.

**Reasons To Accept:**

The authors demonstrate that selecting data which maximizes feature diversity combined with low-rank matrix completion consistently outperforms under different data budgets simple baselines such as random data selection or candidate selection.

**Reasons To Reject:**

The targeted scope is too narrow, even for a short paper: entity disambiguation, a simplified entity linking task, covering a single model applied on a single dataset.

**Reproducibility:**

3: Could reproduce the results with some difficulty. The settings of parameters are underspecified or subjectively determined; the training/evaluation data are not widely available.

**Reviewer Confidence:**

3: Pretty sure, but there's a chance I missed something. Although I have a good feel for this area in general, I did not carefully check the paper's details, e.g., the math, experimental design, or novelty.

**Typos Grammar Style And Presentation Improvements:**

For reproducibility you should state the hyperparameters you considered and also ended up selecting.

---

> ### Author Rebuttal · Authors · 2023-08-28
>
> We thank the reviewer for the review, questions and suggestions, especially the technical suggestions that are very interesting to explore.
>
> - To begin with, let us clarify our perspective on entity disambiguation; we see it as a component, instead of a simplified version, of entity linking. To perform entity linking, one must first recognize the entities and generate candidates before proceeding to disambiguation. While it is possible to learn these tasks jointly, we can also leverage the latest state-of-the-art NER models and combine it with the disambiguation component to create a complete entity linker. It is worth noting that in entity linking studies, the model is usually expected to abstain when the correct entity is not in the candidate list, while in entity disambiguation studies, it is assumed that the correct entity is already in the candidate list. Nevertheless, one can always add logic to the model to allow it to abstain, for example by having a minimum confidence threshold.
>
> - This paper is meant to introduce a new way of looking into a bilinear model and how we can exploit it for sampling. The experiments were done to illustrate that. We agree that it would be interesting to add more experiments on more datasets to strengthen the support of the observations, especially considering that the strategies can be used for other tasks as well. This is a good direction for future work, where we can show how to adopt these strategies to other tasks and datasets. This being said, we believe that the technical insights that we provide make a useful contribution on its own. To the best of our knowledge, we are the first ones to propose a solution for the cold-start setting in the context of bilinear models.
>
> - More excitingly, in light of the current popularity of transformer models, we can think of ways to apply these strategies to the attention matrix. Although it is not trivial because the matrix is positional, we can think of a more general model that includes a lexicalized attention (i.e. a matrix with parameters associated with words from a dictionary rather than positions in the sequence). It is not hard to see how a lexicalized attention matrix can be thought of as weights of a bilinear scoring function. Therefore, it should be possible to extend our sampling and model correction ideas to work on this setting. While this is outside the scope of this study, we believe its publication might open the door for contributions that exploit our proposed ideas in the context of other models.
>
> - We use k-means as it has been used in previous active learning studies. However, we see no reason not to use the clustering method that the reviewer suggested. While exploring different clustering methods is out of the scope of our study, this can make an interesting expansion for our future study.
>
> - With regards to the question of why we settled on a bilinear entity disambiguation model. The truth is that we first developed the idea for cold-start sampling in bilinear models and then we picked a task for which a bilinear model is a natural choice. Bilinear models are natural for entity disambiguation because both the mention context and the KB entity can be embedded in a feature space. Furthermore, while this might seem less obvious, bilinear models for entity disambiguation have also the advantage that they can naturally handle the zero-shot prediction (i.e. making predictions at test time that involve target entities that were not seen in the training data). This is because as long as we have the KB entry for an entity, we can generate its feature representation and compute its compatibility score with a given mention context.
>
> - To the best of our knowledge, there has been no prior work on sampling strategies on a cold start scenario (we don’t assume the existence of a prior model) for entity disambiguation. There have been other studies on entity disambiguation that do not focus on sampling, and in the case of a strictly local model (where there is no additional global document information), results from a bilinear model are competitive.
>
> - The goal of our paper is to introduce a new way of looking into a bilinear model and how we can exploit it for sampling. We think it’s important and timely to introduce this idea now. With the popularity of transformer models with their attention matrices, it is not hard to see that at some level these models also exploit bilinear components.
>
> - A very good point regarding the hyperparameters, we agree that it would improve the reproducibility of the study to include these, we will add all the details in the appendix.

---

### Official Review · Reviewer_d5Yd · 2023-08-05

**Soundness:** 4

**Excitement:**

3: Ambivalent: It has merits (e.g., it reports state-of-the-art results, the idea is nice), but there are key weaknesses (e.g., it describes incremental work), and it can significantly benefit from another round of revision. However, I won't object to accepting it if my co-reviewers champion it.

**Paper Topic And Main Contributions:**

A strategy to improve the generalization performance of bilinear tensor models trained under tight annotation budgets. They combine low-rank matrix completion of a lexical attention matrix and feature diversity sampling.

**Questions For The Authors:**


Section 4.1 is somewhat confusing. It is not easy to follow the connection between collaborative filtering and the proposed approach. Could the authors clarify it a bit?

What is the difference between "testa" and "testb"?

Could the authors provide more clues about the implications of cross-product versus dense diversity strategies?



**Reasons To Accept:**


- The goal is well defined.
- The description of the proposed approach is easy to follow.
- The experiments are not very exhaustive but sufficient to demonstrate the effectiveness of the proposal.

**Reasons To Reject:**

- The state of the art is somewhat short.
- There are many design decisions not entirely justified.
- The experiments do not cover too many aspects.



**Reproducibility:**

4: Could mostly reproduce the results, but there may be some variation because of sample variance or minor variations in their interpretation of the protocol or method.

**Reviewer Confidence:**

2: Willing to defend my evaluation, but it is fairly likely that I missed some details, didn't understand some central points, or can't be sure about the novelty of the work.

---

> ### Author Rebuttal · Authors · 2023-08-28
>
> We thank the reviewer for the review, suggestions, and questions that will help us improve the paper.
>
> - The state of the art is somewhat short because we tried to compress it due to the space limit. Since this paper is quite technical, we focus on writing the technical aspects. Moreover, to the best of our knowledge, there have been no other studies on designing better sampling strategies on a cold start scenario (we don’t assume the existence of a prior model) for entity disambiguation. We have added state-of-the-art related tasks, including those published in conferences outside NLP. Following your concern, we will describe the related studies in more detail, and move some technical explanations to the appendix.
>
> - We agree that some design decisions might benefit from more explanations. Although the experiments might not look like they cover too many aspects, we tried to be selective and attempted to study the most relevant ones. This being said, we see how the paper will benefit from a more qualitative analysis of the results. We have indeed performed a qualitative analysis of the tensor features that are completed in the low-rank correction step but we omitted the section at the end because of space limitations.  This qualitative analysis can shed light on what the model is doing and provide better intuitions for some of our modeling decisions. We will include such analyses in the appendix.
>
> - In section 4.1, we use the analogy of collaborative filtering because we assume it is a more familiar example for the readers. The bilinear model represents two objects in a feature space and computes their compatibility, just as collaborative filtering (e.g. computing compatibility between user vs. movie). Related to the previous point, some qualitative analysis and examples in the appendix might help illustrate how the bilinear model learns sub-classes of words - analogous to sub-classes of users and movies in the collaborative filtering example above. For instance, in AIDA, there are some entities of sports teams with many training examples while other sports team entities have very few examples. In the analogy of user vs. movie collaborative filtering, we can think of these sports team entities as users, and the low-rank approach will “group” users that behave similarly and propagate features to those with fewer examples. We will include these intuitive examples and analysis from the experiment in the appendix to better illustrate what is happening.
>
> - “Testa” and “Testb” are standard partitions in the AIDA data, which is a standard benchmark for entity disambiguation, and thus we omitted the details. However, we agree that we can add these details to the appendix for clarity. Originally, testa is a validation partition, and testb is a test partition. But in our scenario, it is unrealistic to assume that we have access to a large validation partition since we only have a small budget for annotation. We create a small validation set from both testa and testb instead. It has also been well documented in previous studies that due to some difference in the distribution, testa seems to be an “easier” partition (models trained on the standard train set will perform better in testa) than testb. So, we test on both to see if the strategies will behave differently depending on how “hard” the data is.
>
> - Cross-product and dense diversity are both sampling strategies that try to maximize the diversity of the annotated data. The cross-product approach does this by covering as many features as possible, while the dense diversity strategy is trying to cover as many clusters in the data as possible. The implication is that if the feature space is homogenous, there is no natural clustering, and the result of the two strategies will be similar. Otherwise, if the features are highly clustered, we might gain by doing the clustering strategy because we will focus on dense areas of the feature space. On the other hand, it is also possible that the cluster space does not reflect the underlying structure of the feature space and will not help the sampling. We can see that the clustering diversity strategy requires stronger assumptions to work, but this is an obvious strategy that is quite popular in similar scenarios, so we considered both options in our study.

---

### Official Review · Reviewer_U5Tk · 2023-08-12

**Soundness:** 3

**Excitement:**

3: Ambivalent: It has merits (e.g., it reports state-of-the-art results, the idea is nice), but there are key weaknesses (e.g., it describes incremental work), and it can significantly benefit from another round of revision. However, I won't object to accepting it if my co-reviewers champion it.

**Paper Topic And Main Contributions:**


The paper introduces a novel approach for training entity disambiguation models with limited labeled data by combining feature diversity sampling and low-rank matrix completion within a tensor model framework. The method aims to improve generalization performance under tight annotation budgets. Experiments on the AIDA dataset demonstrate that the proposed approach, particularly when integrating low-rank correction, significantly reduces the required labeled data while achieving competitive accuracy. The study contributes valuable insights for effectively addressing the challenge of entity disambiguation in resource-constrained settings.

**Reasons To Accept:**


- The paper addresses a relevant and practical problem in NLP: training entity disambiguation models with limited labeling budgets.
- The paper provides a clear and detailed explanation of the proposed method, including the formulation, algorithms, and reasoning behind the approach.
- Experimental results are presented with a clear comparison of different sampling strategies and the impact of low-rank correction on model performance.

**Reasons To Reject:**


- The paper could benefit from providing more details about the specific datasets used for experiments, including their characteristics and properties.
- While the paper mentions the benefits of the proposed approach, a more qualitative or interpretative analysis of the results could enhance the understanding of why certain methods work better.
- The paper could include a more in-depth comparison with other related works, discussing the strengths and weaknesses of existing approaches and how the proposed method compares.
- Consider discussing potential practical implications of the findings, such as how the proposed method could be applied to real-world applications or scenarios beyond the scope of the current study.


**Reproducibility:**

5: Could easily reproduce the results.

**Reviewer Confidence:**

2: Willing to defend my evaluation, but it is fairly likely that I missed some details, didn't understand some central points, or can't be sure about the novelty of the work.

---

> ### Author Rebuttal · Authors · 2023-08-28
>
> We thank the reviewer for all the suggestions to improve the paper and for highlighting that our study contributes valuable insights for addressing a relevant and practical problem of training a model in resource-constrained settings.
>
> - Due to space limitations and given the fact that AIDA is a well known dataset for entity linking, we omitted a more detailed description. But we understand that some readers might not be familiar with the dataset and we will add all the details in the appendix.
>
> - We agree that qualitative and interpretative analysis would be interesting for this study. In fact, we have done some qualitative analysis of the tensor-features that are completed by the low-rank correction. Unfortunately,  we decided to omit that section at the end due to the space limit. Because the contribution is quite technical we gave priority to explaining the technical details that are necessary to understand the work. Now we realize that the qualitative analysis could be added to the appendix and we will do so in the final version. It is indeed quite interesting because it shows that the matrix completion step is implicitly learning latent classes of features.
>
> - Unfortunately, this paper lacks comparisons with previous studies because to the best of our knowledge, there have been no other studies on sampling strategies for the cold start scenario (we don’t assume the existence of a prior model) for entity disambiguation. Due to this reason, we tried to be rigorous and create strong baselines, taking ideas from other related tasks and domains.
>
> - We realize now that the implications of our contribution might not be obvious. We will highlight here the main implications and try to fit them in the main content in the final version.
>
>    - The main practical implication of our work is that we can use the proposed sampling strategies to train an entity linker more efficiently when we have a new or domain-specific knowledge base, and we need to start from zero, which is a realistic and very frequent scenario in the industry. The sampling strategies can be used to obtain a better model with fewer annotations, this is important because there are many scenarios in which obtaining annotations can be very expensive. For example, as we mentioned in the introduction, a defense research analyst might be interested in mapping mentions of military equipment appearing in military research articles to a target knowledge base describing emergent defense technologies. In this case, we must focus the annotation effort on the most informative examples in the corpus.
>
>    - Furthermore, we believe that the technical implications of our contribution go beyond entity linking. The sampling ideas and model correction strategies that we propose can be useful for any task that can benefit from a bilinear component.  This includes a wide range of tasks. Essentially, any prediction task that needs to compute a compatibility score between an input and an output and that it does so by first projecting each of them to some feature space.
>
>    - More excitingly, in light of the current popularity of transformer models, we can think of ways to apply these strategies to the attention matrix. Although it is not trivial because the matrix is positional, we can think of a more general model that includes a lexicalized attention (i.e. a matrix with parameters associated with words from a dictionary rather than positions in the sequence). It is not hard to see how a lexicalized attention matrix can be thought of as weights of a bilinear scoring function. Therefore, it should be possible to extend our sampling and model correction ideas to work on this setting. While this is outside the scope of this study, we believe its publication might open the door for contributions that exploit our proposed ideas in the context of other models.

---

### Meta-Review · Area_Chair_mXQr · 2023-09-15

**Recommendation:** 3

**Metareview:**

The paper innovatively combines feature diversity sampling, low-rank matrix completion in tensor models, and improving entity disambiguation with limited labeled data.

The proposed method for entity linking is clearly explained, and the experimental results support the paper's claims. However, the experimental section primarily focuses on a specific dataset, which somewhat limits its ability to fully demonstrate the method's applicability in different scenarios.

---

### Decision · Program_Chairs · 2023-10-07

**Decision:**

Accept-Findings

**Comment:**

The paper innovatively combines feature diversity sampling, low-rank matrix completion in tensor models, and improving entity disambiguation with limited labeled data.

The proposed method for entity linking is clearly explained, and the experimental results support the paper's claims. However, the experimental section primarily focuses on a specific dataset, which somewhat limits its ability to fully demonstrate the method's applicability in different scenarios.